# Anti-Quorum Sensing Activity of Probiotics: The Mechanism and Role in Food and Gut Health

**DOI:** 10.3390/microorganisms11030793

**Published:** 2023-03-20

**Authors:** Mohammed Kamal Salman, Jumana Abuqwider, Gianluigi Mauriello

**Affiliations:** Department of Agricultural Sciences, University of Naples Federico II, 80055 Portici, Italy

**Keywords:** quorum sensing, quorum sensing inhibition, quorum quenching, virulence, biofilm, microencapsulation, gut

## Abstract

Background: Quorum sensing (QS) is a cell-to-cell communication mechanism that occurs between inter- and intra-bacterial species and is regulated by signaling molecules called autoinducers (AIs). It has been suggested that probiotics can exert a QS inhibitory effect through their metabolites. Purpose: To provide an overview of (1) the anti-QS activity of probiotics and its mechanism against foodborne pathogenic and spoilage bacteria; (2) the potential role of the QS of probiotics in gut health; and (3) the impact of microencapsulation on QS. Results: *Lactobacillus* species have been extensively studied for their anti-QS activity and have been found to effectively disrupt QS in vitro. However, their effectiveness in a food matrix is yet to be determined as they interfere with the AI receptor or its synthesis. QS plays an important role in both the biofilm formation of probiotics and pathogenic bacteria. Moreover, in vitro and animal studies have shown that QS molecules can modulate cytokine responses and gut dysbiosis and maintain intestinal barrier function. In this scenario, microencapsulation was found to enhance AI activity. However, its impact on the anti-QS activity of probiotics and its underlying mechanism remains unclear. Conclusions: Probiotics are potential candidates to block QS activity in foodborne pathogenic and food spoilage bacteria. Microencapsulation increases QS efficacy. However, more research is still needed for the identification of the QS inhibitory metabolites from probiotics and for the elucidation of the anti-QS mechanism of probiotics (microcapsules and free cells) in food and the human gut.

## 1. Introduction

Quorum sensing (QS) is the mechanism that bacterial cells use for inter- and intra-species communication after the cell density reaches a specific threshold. This communication is signaled through molecules produced by the cells and named autoinducer (AI) molecules. When these molecules are accumulated at a specific threshold of concentration, they bind to a detector protein that initiates the QS transduction pathway. This ultimately leads to the genes responsible for pathogenicity, spoilage, and biofilm formation being expressed [1]. There are three common QS systems, which are employed by bacteria to regulate gene expression. The first is the *luxRI*-type QS system that is used by most Gram-negative bacteria. This QS system is mediated through acyl homoserine lactone (AHL) molecules, which are synthesized by the *luxI* gene as AIs. These then bind to the LuxR-type protein encoded by the *luxR* gene to form an AHL–LuxR complex, which activates the target gene expression [2]. The second is a two-component QS signaling system that is used by most Gram-positive species. This QS system is peptide-mediated and termed an accessory gene regulator (Agr) or Agr-like QS system. The AI peptide (AIP) is produced by *agrD* and processed by *agrB* for transportation into the *agrCA* pathway (two-component signaling cascade), in which a membrane-bound histidine kinase is synthesized by *agrC* to detect AIP. This initiates a series of phosphorylation that ultimately leads to the transcriptional regulator kinase (synthesized by *agrA*) being phosphorylated, which triggers the expression of the target genes [3]. A third major QS system is the AI-2/*LuxS* homolog that is shown both in Gram-negative and Gram-positive bacteria. AI-2 is synthesized by *luxS*, which then binds to the AI-2 receptor according to the type of bacteria [4]. A fourth type of autoinducer is AI-3, which was discovered in *Escherichia coli* as a regulator of virulence factors. This type of AI is produced through a series of reactions induced by threonine dehydrogenase and aminoacyl-tRNA synthetases [4].

Microbiome–microbiome crosstalk in the food and gut through QS plays a role in food quality and gut health, which is important for the food industry and medical therapy [5,6]. QS is involved in microbial pathogenesis and biofilm formation in food products, which is deemed a concern for public health and food safety. For example, milk can be contaminated with pathogenic Gram-positive bacteria such as *Listeria monocytogenes* that use peptides (i.e., AIP) as QS signaling molecules to trigger gene expression for biofilm formation [7]. Biofilm formation and microbial enzymatic activity (e.g., pectinolytic, proteolytic, and lipolytic) are factors of food spoilage, which can be regulated by QS. As a matter of fact, *Pseudomonas* and *Enterobacteriaceae* can spoil milk by the proteolytic activity, whose expression is regulated by the *luxRI*-type QS system [8,9]. Spoilage in meat and vegetables is also caused by Gram-negatives, where homoserine lactone (HSL) acts as a signaling molecule to QS [8]. Several studies have shown the positive effect of probiotics on food safety and quality through anti-QS activity [10,11,12].

In addition, the regulation of intestinal homeostasis and inflammation is also controlled by the QS of gut microbiota. It is worth noting that pathogenic bacteria can boost their survival in the gut by forming biofilms, which are regulated by QS [13]. Hence, QS acts as a mediator between gut microbiota and intestinal health [13]. Interestingly, QS inhibition could be one of the mechanisms by which probiotics modulate the gut microbiome and alleviate the harmful health effects of pathogenic bacteria. With QS playing a dual role in both food quality and gut health, disrupting QS could improve food quality and promote gut health. This could potentially be achieved by using probiotics as QS inhibitors (QSIs). Furthermore, the use of probiotics as QSIs may also be considered an alternative therapy to combat antimicrobial bacterial resistance [14]. Many QSIs from different sources were investigated against foodborne pathogenic and food spoilage bacteria. Plant sources of phytochemicals and bioactive compounds, bee products, and bacteria were found effective in disrupting the QS pathway and biofilm formation by spoilage and pathogenic Gram-negative and Gram-positive microorganisms [8,11,15]. As well described in the literature, the QS inhibition could be classified in one of the following mechanisms: (1) repressing the gene encoding the AI synthase; (2) inactivation of AI signals through quorum quenching (QQ) enzymes, (e.g., hydrolysis of the lactone bond in AHL); and (3) interfering with signal receptors in QS networks [14].

There is a growing body of research on the role and mechanism of QS in regulating bacterial biofilm formation and pathogenesis. The food industry is increasingly interested in the use of probiotics for food functionalization and bio-preservation. Therefore, this review has a dual purpose. Firstly, it aims to provide an up-to-date overview of the anti-QS activity of probiotics against foodborne pathogens and spoilage bacteria. Secondly, it aims to offer insights into future perspectives for the potential use of probiotics in food products to enhance food quality and gut health by inhibiting QS. The results of this review will serve as a foundation for further investigations to fill the knowledge gap regarding the role of QS in food quality and human health.

## 2. Mechanism of QS Inhibition 

In the last decade, there has been growing attention to unveil the mechanism that probiotics use to alter QS activity in foodborne pathogenic and spoilage bacteria. Probiotics, in particular lactic acid bacteria (LAB), are the most studied microorganisms as a promising source of QSIs. The interference with the *LuxRI*-type, Agr-like, and AI-2/*LuxS*-type QS systems would result in the repression of virulence genes responsible for pathogenicity, food spoilage, and biofilm formation. In the two sections below, the mechanism of QS disruption in foodborne pathogenic and spoilage bacteria is discussed in detail.

### 2.1. Probiotics as QSIs in Foodborne Pathogenic Bacteria

It is well reported that LAB have a bactericidal effect through the secretion of organic acids, hydrogen peroxide, and bacteriocins, which affect the growth or the survival of pathogenic bacteria [16]. Interestingly, the detrimental effects of foodborne pathogenic bacteria can be reversed by disrupting the QS mechanism, without their growth/survival necessarily being impacted [8]. Numerous probiotics are demonstrated as QSIs through interference with the QS signaling pathway of the target bacteria [11]. LAB are the most studied QSI bacteria against a range of common pathogenic bacteria, which rely on QS for biofilm formation and the expression of virulence factors [11]. 

In Table 1, potential probiotic LAB and non-LAB are described as QSIs. Generally, the interference of QSI with one of the QS pathway components would lead to the downregulation of QS-related genes and the inactivation of QS.

Various strains of *Lb. acidophilus* exhibited anti-QS activity towards *C. difficile, Staph. Aureus*, and *E. coli* [18,24,27] through the inhibition of AI-2 production or downregulation of biofilm-related genes without an influence on AI-2 synthesis. However, the anti-QS compounds in these studies are not well identified. Other similar studies found that the biofilm-related genes in *E. coli* and *P. aeruginosa* were under-expressed by the activity of EPS and DKP produced by the strains of *Lb. acidophilus* A4 and CRL 730, respectively [25,29]. The anti-biofilm and anti-QS activity of probiotics against *Salmonella* was also observed, but the QSIs were not identified [35,36,37,38,39]. In these studies, EPS might be produced by the lactobacillus species as the QSI. This was suggested according to the study of Xu et al. (2020) [40], which found that the biofilm formation of *S*. Typhimurium was repressed by the action of EPS synthesized by *Lb. coryniformis* NA-3. Even though not investigated, EPS from the *Lactobacillus* species could play a role in the interference with the QS in *Salmonella* through the downregulation of the genes encoding the QS regulator proteins [35,36,38,39,40]. Unfortunately, the mechanisms behind the QS inhibition against *E. coli* and *Salmonella* were not well identified. Hypothetically, the QSIs antagonize the QS regulator proteins, which modulate the expression of virulence genes. This was proposed according to the outcomes in other studies that used molecular docking and bioinformatic methods. The chemically produced QSIs interact with the amino acid sequence of the cognate regulator proteins (LsrR and SdiA), which, in part, mediates QS in *Salmonella* and *E. coli* [54,55]. The binding of phosphorylated AI-2 (AI-2-P) with its cognate protein LsrR leads to the inactivation of LsrR, which regulates transcription of the *lsr* operon that controls the cell uptake of AI-2 and pathogenicity [56]. Therefore, QSIs antagonize the LsrR protein by blocking AI-2-P binding, making active LsrR that leads to the repression of the *lsr* operon. Another potential QSI mechanism is the binding of the QSI with SdiA amino acids. This interaction will prevent AHL by binding the SdiA receptor, which works as a transcriptional regulator for QS-related virulence genes [56]. 

*Staph. aureus* is one of the common causative factors for foodborne pathogenesis and infectious mortality globally. It is expected that there is a 10 to 30% mortality rate of SAB (*Staph. aureus* bacteremia) cases, and this is exacerbated by the ability of *Staph. aureus* to develop resistance to antibiotics [57]. In comparison with QS in other bacterial species, *Staph. aureus* encompasses 13 QS pathways, each one associated with a different virulence mechanism [58]. The blocking of QS in *Staph. aureus* could be a surrogate of antibiotic treatment to control virulence expression and infection treatment. Several studies found that probiotics or their metabolites have a QS inhibitory effect against *Staph. aureus*. As a matter of fact, cyclo-dipeptides produced by *Lb. reuteri* showed repression activity on *Staph. aureus* pathogenicity elements, such as the Agr system, *tst* gene, and *SaeRS* QS system [20]. A cross-sectional study revealed an *agr*-inhibitory effect via *B. subtilis*-produced lipopeptide (fengycin) [21]. In this study, 200 nasal swapes and fecal samples were collected from a Thai population living in different areas and were analyzed for *Staph. aureus* and *B. subtilis* colonization. There was a positive relationship between the presence of *B. subtilis* and the absence of *Staph. aureus.* Furthermore, *agr* activity declined when the *B. subtilis* culture filtrate was incubated with the *Staph. aureus* reporter strain, and the synthesis of virulence factors (phenol-soluble modulins, α-toxin, and Panton–Valentine leucocidin) was also suppressed [21]. Accordingly, the two-component signaling (*fsr*) in *E. faecalis* was disrupted by the inhibitory effect of *B. subtilis* ZK3814-secreted lipopeptides (surfactin and fengycin). The result was that the expression of proteolytic activity-related genes (*gelE*/*sprE*) was quenched. 

Other anti-QS unraveled compounds are the biosurfactants synthesized by *Lb. helveticus* and *P. pentosaceus*, which were shown to reduce biofilm formation by *B. subtilis, P. aeruginosa*, *Staph. aureus*, and *E. coli* [22,42]. While the anti-QS activity of the biosurfactant from *P. pentosaceus* was not investigated in-depth [42], Jiang et al. (2019) [22] found that the biosurfactant from *Lb. helveticus* reduced in *Staph. aureus* the expression of *dltB*, which is a regulatory gene in the two-component signaling system *GraRS* that controls the resistance to cationic antimicrobial peptides (CAMB), which is a defense system that a host employs against microbial infection. The response regulator genes *agrA* and *sarA* that are involved in the regulation of biofilm formation were also repressed. Interestingly, the in vivo experiment showed that the anti-biofilm activity also reduced the hemolytic activity of *Staph. aureus*. Peculiarly, the AI-2 release was associated with the inhibition of biofilm formation as found by Ahn et al. (2021) [23]. LPA produced from *Lb. plantarum* KCTC10887BP reduced the expression of biofilm formation by the induction of AI-2 release. This was confirmed in another study [59], which reported an upregulation of biofilm formation in *Staph. aureus* RN6390B with mutation in the *luxS* gene. AI-2 release results in the transcription of the *icaR* gene, which works as a repressor for the *ica* operon, and consequently, the expression of the *icaA* gene that is involved in biofilm formation is deactivated. 

Having established the anti-biofilm activity of probiotics, the tooth decay causing-bacterium *Str. mutans* relies on biofilm formation for adhesion. Several studies show that the anti-QS activity of probiotics is related to the repression of biofilm-related genes and the inhibition of EPS production, which is an important matrix of biofilm formed by *Str. mutans*. Biosurfactants and a novel compound (iminosugar) were identified as QSIs produced by *Lb. rhamnosus* and *Lb. paragasseri* MJM60645, respectively [47,48,49,50,51]. *Lb. plantarum*-secreted LPA exerted anti-biofilm activity against *Str. mutans*, *E. faecalis*, and *Str. gordonii* [52]. This anti-biofilm activity could be due to the same inductive effect of LPA on AI-2 release that was observed by Yu et al. (2012) [59], albeit the QSI mechanism is not described. 

*P. aeruginosa* is a pathogenic bacterium that causes infections in humans, especially in immunocompromised populations. The treatment of infections with *P. aeruginosa* is cumbersome due to its resilience to various stress conditions and antibiotic administration and its ability to produce diverse types of EPS matrixes for biofilm formation and cell adhesion [60,61]. There are three QS systems (*las*, *rhl*, and *pqs*) in *P. aeruginosa*, which are activated for the expression of biofilm formation and virulence factors. *LasI*, *rhlI*, and *pqsABCDH* are synthase genes of the autoinducers N-(3-oxododecanoyl)-HSL, N-butyryl-HSL (C4-HSL), and 2-heptyl-3-hydroxy-4-quinolone (PQS), respectively [62]. Extensive research has been conducted to investigate the QSI by probiotics and their metabolites in vitro. QS was blocked through the degradation of QS signal molecules (AHL) by various species of *B.*
*paralicheniformis*. Lactonase and acylase were found to degrade the lactone and acyl rings in the QS signal AHL, respectively [33]. *Lb. brevis* showed anti-QS enzymatic activity by unknown AHL-degrading enzymes [28]. It is noteworthy that as lactonase and acylase only degrade AHLs, these enzymes are only efficient in Gram-negative bacteria, which involve AHL-based QS pathways. Another route of QS inhibition against *P.*
*aeruginosa* is the antagonization of the receptor proteins. Stigmatellin Y from *B. subtilis BR4* interfered with the PQS QS systems by binding with the PqsR protein in competition with the PQS signal molecule and disruption of AHL-signal transduction [32]. Lyophilized postbiotics from *Lb. casei* disrupted the *rhl* QS system and downregulated the genes for autoinducer synthase and its cognate regulatory protein, potentially, due to the action of organic acids [31]. When the CFS of *Lb. rhamnosus* GG was incubated with *P. aeruginosa* PAO1, there was a decrease in AHL activity and levels [30]. In this study, the authors state that the CFS has inhibited AHL production, potentially by reducing the expression of the AHL-producing gene (*lasI*), albeit this was not evaluated. Perhaps, *Lb. rhamnosus* GG reduced the AHL activity through the newly revealed enzyme (acyl-hydrolase); however, this still needs further investigation [63].

Bacteriocins are antimicrobial peptides that are usually secreted from lactic acid bacteria that kill or inhibit the growth of the target pathogenic or spoilage microorganisms. They have lately been linked with the QS mechanism. In Rizzello et al. (2014) [11], QS in *Lb. plantarum* C2 was involved in the production of plantaricin. Reuterin and lactocin released from *Lb. reuteri* LR 21 and *Lb. curvatus* CRL1579, respectively, acted as QSIs [17,49]. Reuterin reduced the expression of the *luxS* and *agrB* genes in *C. perfringens* 13124, which subsequently prevented toxin production by repressing the toxin-producing genes (*cpa* and *pfo*). The QSI mechanism of lactocin is not yet demonstrated. However, in Hossain et al. (2021) [56], the expression of the biofilm-regulatory genes of *L. monocytogenes* ((*flaA*, *fbp*, *agrA*, *prfA*, and *hlyA*) was inhibited as an effect of the postbiotics from *Lb. curvatus* B.67 and *Lb. plantarum* M.2. Very recently, *Lb. curvatus* B67 was reported to produce quercetin that shows anti-biofilm activity against *Listeria monocytogenes* [64]. Reasonably, a QS inhibition activity of quercetin could be hypothesized.

Apart from the QSI mechanism discussed above, it is suggested the QS pathway for biofilm formation could be disrupted by the inhibition of proton motive forces (PMFs) and efflux pumps. PMFs were associated with the release of AIs from cells, and the inactivation of PMFs would lead to the intracellular accumulation of autoinducers and thus, the downregulation of QS-regulatory genes [65]. This was also confirmed when QS peptide export was blocked due to the disruption of PMFs in *Str*. *pneumoniae* [66]. In addition, in Sutyak et al. (2011) [67], the inhibitory effect of subtilosin on PMF in *Gordonia vaginalis* was reported. Therefore, the anti-biofilm activity of *B. subtilis* KATMIRA1933 against *L. monocytogenes* could be potentially attributed to the inhibitory effect of subtilosin on PMFs [37].

In summary, probiotics could exhibit QS inhibitory effects by producing and secreting QSIs. These are reported as lipopeptides (such as fengycin and surfactin), lipoproteins (biosurfactants), organic compounds, cyclic peptides, lipoteichoic acid, exopolysaccharides, bacteriocins (such as reuterin and lactocin), and AHL-lytic enzymes. However, in some studies, there was a gap in the identification of QSI compounds or the elaboration of the QS inhibition mechanism when anti-QS activity was shown.

Despite the anti-QS effect of the CFS of probiotics/postbiotics against foodborne pathogens, it is not known whether the same anti-QS activity can be observed if probiotics are investigated as cell cultures. Furthermore, the extensive research on QSI has been conducted in in vitro experiments. In fact, the outcomes of these studies do not necessarily replicate the context in real foods. The chemical composition of food (proteins, lipids, and carbohydrates) might affect the viability and functional properties of probiotics [68].

### 2.2. Potential Role of Probiotics in QS Inhibition in Food Spoilage Bacteria

Food spoilage is a consequence of physical changes, such as texture change due to drying, chemical reactions, such as oxidation, and microbial activity. Microbial-induced food spoilage can reduce the shelf-life of food by altering its sensory qualities, such as texture, flavor, and taste, due to enzymatic activity (proteolytic, lipolytic, and pectinolytic) and biofilm formation [69]. QS molecules have been detected in different types of food products, including meat, milk, and vegetables, and their presence has been suggested to be involved in microbial food spoilage [8]. The growth of psychrotrophic Gram-negative bacteria, usually isolated from raw milk (e.g., Pseudomonas and Serratia), is associated with biofilm formation and the production of extracellular enzymes that are mediated by the luxI/AHL-type QS system [70]. This has also been confirmed in a transcriptome analysis study that evaluated the effect of exogenous AHLs on the virulence activity of Pseudomonas azotoformans and Serratia liquefaciens in milk. It was observed that the addition of AHLs upregulated the genes responsible for growth, metabolism, and enzymatic activity (protease, lipase, and glycosidase) [71]. Fresh meat contains a complex of several bacterial genera (Enterobacteriaceae, Yersinia, and Pseudomonas) that induce spoilage. In these species, the expression of biofilm formation and virulence factors of spoilage are stimulated by the luxI/AHL-type QS signaling pathways. Proteolytic activity and AHLs with different carbon chain lengths were found in chilled meat products, and this was associated with the growth of Enterobacteriaceae and Pseudomonas. Moreover, the luxS/AI-2-type QS system is known to be implicated by Leuconostoc citreum, a lactic acid bacterium that causes meat spoilage in biofilm formation [9,72,73,74,75]. Post-harvest contamination of vegetables with pectinolytic and proteolytic bacteria could impact the quality of the product and reduce the shelf-life. Pectobacterium carotovorum is one of the common plant pathogenic bacteria that causes soft-rot disease in a wide range of vegetables (potato, onion, tomato, and radish) through enzymatic activity, which is a concern for the agriculture industry and farmers due to the economic loss [76]. The production of pectinolytic and proteolytic enzymes is regulated by QS. It was found that LuxS-mutant Pectobacterium carotovorum reduced the expression of pectate lyase and cellulase in addition to protease enzymes, which demonstrates the role of the luxS/AI-2-type QS system in enzymatic activity expression [77]. Similarly, in LuxS-mutant *Erwinia carotovora*, there was a four- and three-times decrease in the production of pectate lyase and polygalacturonase compared with the wild-type strain, after 6 and 8 h of growth, respectively [78]. In the same species, another study showed the role of the luxI/AHL-type QS system in the rot–spoilage of bean sprouts by proteolytic and pectinolytic activity. However, when the bean sprouts were inoculated with the luxI-mutant strain, the spoilage took longer to occur compared with sprouts contaminated with the wild-type strain [79]. Shewanella baltica is the primary bacterium that causes fish to deteriorate. It also employs the luxS/AI-2-type QS system to activate genes related to spoilage and biofilm formation through DKPs as signaling molecules for QS induction [80]. Overall, QS plays a crucial role in microbial food spoilage. Therefore, disrupting QS signaling in spoilage bacteria could extend the shelf life of fish products by preventing the expression of virulence factors and biofilm formation. A wide range of QSIs, including plant-derived compounds (such as phenol acids), chemicals such as furanone, and QS signal-degrading enzymes, have been demonstrated [8]. This section explores the potential role of probiotics as a source of QSIs.

Notably, QS inhibition against microbial spoilage is a recent focus of research. As shown in Table 2, only five studies have investigated potential probiotics as a source of QSIs. The anti-QS activity of *Bacillus strains* was identified through the QQ activity of lactonase and the antagonistic effect of DKPs on QS receptors. Li et al. (2022a) [81] showed that QS promoted the growth of *Lb. plantarum* ss-128, but they did not investigate its anti-QS activity against spoilage bacteria. Research on the anti-QS activity of lactic acid bacteria against food spoilage bacteria is still lacking.

Spoilage bacteria can cause changes in food quality due to both enzymatic activity and the release of chemical compounds, including volatile organic compounds (VOCs), volatile fatty acids, ethyl esters, aldehydes, and sulfur compounds, which can result in sensory defects such as off-odor, off-flavor, and discoloration [72]. It is unclear, however, whether QS directly influences the expression of these virulence factors.

The interaction between probiotics and spoilage bacteria in food microbiomes has the potential to play a key role in prolonging food shelf-life by modulating the expression of spoilage factors. However, the effect of such an interaction on QS in spoilage bacteria is still not well-researched. If probiotics are found to have an anti-QS effect against spoilage bacteria, this could be a significant step towards replacing chemical preservatives with bio-preservatives and enhancing the efficacy of other preservatives.

## 3. QS, Biofilm Formation, and Gut Health

It is well-known that probiotics provide health benefits for the host gut through several mechanisms. This section briefly discusses the role of probiotics’ QS system as one of these mechanisms, as summarized in Table 3.

The induction or inhibition of QS signaling is associated with the expression or repression of biofilm-related genes, respectively. In fact, biofilm is necessary for pathogenic bacteria to maintain their growth and infect the host environment [94]. The protective effect of probiotics against pathogenic bacteria in the gut could be attributed to their anti-QS and anti-biofilm activity, as shown in Table 1. Additionally, the QS system is involved in the regulation of biofilm formation by probiotics in the host gut [10]. This forms a biological barrier and maintains intestinal immunity against pathogenic bacteria that cause infection [10]. The importance of the AI-2/*luxS*-type QS system in biofilm formation and stress resistance of the probiotics (*Lactobacillus* and *Bifidobacterium*) was observed. The promotion of AI-2 synthesis by the overexpression of *luxS* in *Lb. paraplantarum* L-ZS increased resistance to heat- and bile-salt stress compared with the wild-type strain [95]. Moreover, the biofilm formation of *Lb. paraplantarum* L-ZS and *Bifidobacterium longum* NCC2705 was augmented [95,96].

An emerging field of research highlights the ability of QS molecules to modulate the inflammatory response. One type of AHL QS molecule, 3-oxo-C12:2-HSL, was described as anti-inflammatory through the activation of bitter taste receptors and the inactivation of cell signaling cascades (JAK-STAT signaling pathway, NF-κB signaling, and TNF signaling pathway) that upregulate the inflammatory response [86,97]. Additionally, the presence of 3-oxo-C12:2-HSL in the gut was negatively correlated with inflammatory bowel disease (IBD) incidence, indicating the role of this type of AHL as an anti-inflammatory [87]. Another type of QS molecule, AI-2, was recently found to be able to modulate the inflammatory response and gut microbiota composition in vivo [89,90]. In a study by Thompson et al. (2015) [88], a group of mice was induced into dysbiosis by antibiotic administration and then supplemented with genetically engineered *E. coli* overproducing AI-2. After treatment, the antibiotic effect was reversed, and the abundance of *Firmicutes* was significantly promoted compared to *Bacteroidetes*. The authors of this study propose that AI-2 improved the signaling response among the *Firmicutes*, a recurrent producer of AI-2. In another study [89], a group of early weaning stress-induced piglets was treated with either the wild type (WT) or *luxS* mutant (Δ*luxS*) of *Lb. rhamnosus* GG. The ability of Δ*luxS* to preserve the gut barrier function was significantly lower than the WT. As previously discussed, this is due to the role of AI-2 in biofilm formation, which the Δ*luxS* strain lacks to colonize and adhere in the intestine cells. This postulates the role the AI-2/*luxS*-type QS system has in gut health. Recently, an animal study showed that AI-2 supplementation of mouse pups with necrotizing enterocolitis reduced the expression of pro-inflammatory cytokines and increased the anti-inflammatory cytokine (IL-10) production. Furthermore, AI-2 rebalanced the composition of gut flora. In the mice group treated with AI-2, *Lactobacillus* was increased, while *Helicobacter* and *Clostridum_sensu_stricto_1* were reduced compared with the not-AI-2-treated group [90]. In another study, the butyl-DPD analog (the precursor for AI-2) was able to alleviate the inflammatory response of human gingival keratinocyte cells when incubated with two *Prevotella* species. The interleukins’ (IL-6 and IL-8) expression was reduced [91]. Nevertheless, a cross-sectional study in patients with colorectal cancer (CRC) demonstrated the association of AI-2 secreted from gut microbiota with CRC occurrence [92]. In addition, AI-2 from non-pathogenic *E. coli* mediated the transcription of the cytokine response pathway, which as a result, induced inflammatory interleukin IL-8 through the upregulation of the transcription factor NF-κB pathway that regulates cytokine production [93].

There are chemically synthesized QSIs that can be considered anti-virulence agents for infection treatment. Nevertheless, it is proposed that some species such as *E. coli* and *Salmonella* can develop resistance to these QSIs, which is partially due to the opportunity for bacteria to shift from the QS system to another strategy for virulence expression. In the review study by Escobar-Muciño et al. (2022) [55], the ability of bacteria to develop QSI resistance via enhancing the affinity of QS-receptor proteins to QSIs is discussed. In addition, there are critics that were reported to point out the limitations of QSI use as a therapeutic strategy against infectious pathogens [98]. As discussed above, AI-2 signaling is required for biofilm formation for both probiotics and pathogenic bacteria. The interruption of AI-2 signaling using unselective chemical inhibitors, such as furanone, would also act on probiotic gut microbiota, which would then reduce their resistance to pathogenic microorganisms [98]. Furthermore, in Δ*agr*-mutant *Staph. aureus*, biofilm formation and higher survivability were observed, and in Δ*lasR*-mutant *P. aeruginosa*, higher production of pyocyanin (virulence factor) and pro-inflammatory responses have also been reported. These observations might restrict the efficacy of chemical QSIs [99]. Finally, foodborne pathogenic bacteria could develop resistance to QSIs as shown by studies that revealed the ability of *P. aeruginosa* to develop resistance to antibiotics (carbapenems and azithromycin) that show anti-QS activity [98].

The potentiality to develop resistance to QSI-based therapy is prompting the opening of a new line of investigation to unravel the mechanism of probiotics for use as a substitutive chemical QSI-based therapy against infectious diseases to overcome the microbial QSI resistance. 

It can be deduced that QS is an interplay in host–gut microbiome interaction. The health benefit mechanism of probiotics is either by reducing the expression of QS biofilm-related genes of pathogenic bacteria or by direct impact of QS molecules on intestinal cells. However, the results are still contradictory. In Li et al. (2019) [96], AI-2 was suggested to be involved in the progression of CRC and can be used as a marker, whereas as described by the abovementioned studies [92,93,94,95], there is a positive role of AI-2 in the alleviation of pro-inflammatory responses. Furthermore, the AI-2 used in some of the above studies was chemically synthesized and supplemented exogenously in in vivo studies. It will be intriguing to investigate the effect of the AI-2 secretions from probiotics on gut dysbiosis, inflammatory response, and intestine barrier health in in vivo and human studies. In addition, research on the mechanism of probiotic health benefits in the gut via QS is still lacking. 

## 4. Microencapsulation and QS

Probiotic microencapsulation is used to increase stress resistance and cell viability after exposure to the outside environment, such as during passage in the gastrointestinal tract, processing, and storage [99,100]. Microencapsulation of probiotics also boosts the health benefits of probiotics. An animal study found that microcapsules of *Lb. plantarum* LIP-1 reversed dysbiosis, increased colonization, and reduced lipid concentration in hyperlipidemic rats compared with free cells of LIP-1 [101]. In addition, microencapsulation of *Lb. plantarum* LN66 increased the survival rate under different packaging conditions and simulated a gastrointestinal environment compared with free cells [102]. It is possible that the microencapsulation mode of action on bacterial behavior is partially regulated by QS. During the incubation of microcapsulated cells, bacteria are grown in aggregates with shorter in-between cell distances than in free cells. This conduces to better AI activity and cell-to-cell communication, that is, the QS efficacy is not the same when bacterial cells are free or encapsulated. This was found in the study by Gao et al. (2016a) [103] when *V*. *harveyi* cells were incorporated into microcapsules, and the QS capacity was evaluated in comparison with free cells. The results showed that the *luxS* gene expression was significantly higher in microencapsulated cells than in free cells, despite the cell density being lower in microcapsules after 12 h of incubation. In contrast, another study found that the viable *Lb. plantarum* AB-1 cells were significantly higher in microcapsules compared with free cells after 48 h of growth culturing, but the same outcome of enhanced QS was observed [104]. This might be due to different types of bacteria or the experimental design. Moreover, the size of cell aggregates plays a part in the impact of microencapsulation on QS mediated by the spatiality effect. Large cell aggregates demonstrated more effective QS than small cell aggregates (Gao et al., 2016b) [105]. Prominently, bacterial QS can be also regulated by manipulation of the microenvironment inside the microcapsules (diameter, core state, and alginate concentration). Alginate concentration and microcapsule diameter are positively correlated with AI activity and, thus, stronger QS capacity. Furthermore, when the core state of a microcapsule is solid, the size of the cell aggregate is larger than in a liquid core state. As mentioned previously, the QS is more enhanced in large cell aggregates than in small cell aggregates. The underlying mechanism of these microenvironment parameter effects is that the mass transfer of AIs inside the microcapsule is reduced in response to the increase in diameters and alginate concentration and the solidified core state. This makes the AIs restricted inside the microcapsules, which is, in turn, how the interaction of the AIs with cells is facilitated [106]. Considering the impact of microencapsulation on QS efficacy, it would be worth investigating whether the anti-QS and anti-biofilm effect of probiotics can be improved by microencapsulation. Until now, there is only one study that investigated the anti-biofilm activity of *Lb. rhamnosus* GG microcapsules in co-culture with *E. coli* [13]. The results indicated that *E. coli* QS-related genes encoding biofilm formation (*lsrK* and *luxS*) were significantly downregulated as a result of the anti-QS activity of the microencapsulated cells of *Lb. rhamnosus* GG. Interestingly, microcapsules have not only been able to cause a remarkable decline in biofilm formation but also to erase mature biofilm. In a study conducted by Song et al. (2021) [107], the anti-biofilm properties of microencapsulated *Lb. rhamnosus* GG were confirmed through a proteomic quantitative analysis of a co-culture of *E. coli* and the microcapsules. The microcapsules disrupted the metabolic pathway of *E. coli*, resulting in a reduction of biofilm formation and led to decreased expression of genes related to biofilm formation and stress resistance (*dnaK* and *bamE*), although it is unclear whether this was due to QS inhibition. However, the efficacy of the anti-QS activity of the microcapsules was not compared to that of free cells, and it is important to determine to what extent the microcapsules enhance anti-QS activity. Furthermore, the QQ compounds were not identified in these studies.

## 5. Concluding Remarks and Future Perspectives

QS serves a crucial and integral function in bacterial behavior in response to stress and environmental conditions. This makes QS a recent focus of research to maintain food safety and quality. Probiotics can act as QSIs by interfering with the QS activity of the target bacteria through the secretion of metabolites. The microencapsulation of probiotics is a promising strategy to enhance the anti-QS activity of probiotics. However, most of the studies in the literature are in vitro, with little focus on replicating the outcomes in the food matrix. Probiotics, through anti-QS activity, could play a potential role in gut health through the modulation of pro-inflammatory responses and gut microbiota. Taken together, the future research hotspot could be focused on the investigation of the following: (1) the anti-QS activity of probiotics, identification of anti-QS compounds, and their mechanism against foodborne pathogenic and spoilage bacteria in food products; (2) the impact of microencapsulation on the anti-QS activity of probiotics and its mechanism in the food matrix; and (3) the potential role of probiotic QS in the modulation of inflammatory responses and gut microbiota in human studies.

## Figures and Tables

**Table 1 microorganisms-11-00793-t001:** Probiotics involved in the inhibition of QS mechanisms of foodborne pathogenic bacteria.

Microorganism	QSI	Target	Type of Study	Mechanism	Reference
*Lb. reuteri* LR 21	Reuterin	*C. perfringens* 13124	In vitro	Repression of toxins-producing genes (*cpa* and *pfo*) and *agrB* and *luxS.*	[17]
* Lb. acidophilus * GP1B	CE/CFS	* C. difficile * * (ribotype 027) *	In vitro	Inhibition of AI-2 production and downregulation of *luxS* and *tcdA*, *tcdB*, and *txeR* (virulence genes). Growth inhibition of *C. difficile* in the colon.	[18]
* Lb. ** fermentum * Lim2	Inactivated CE	* C. difficile * 027	In vitro	Anti AI-2 activity due to repression of *lux* gene. Expression of virulence genes also reduced. QSIs are not measured.	[19]
*Lb. reuteri* RC-14	CFS	*Staph. aureus* MN8	In vitro	Cyclo-dipeptides inhibited the expression of *agr* and *tst* genes as well as disrupting *saeRS* system.	[20]
*B. subtilis*	Fengycin	*Staph. aureus*	Cross-sectional analysis (Thai population)	Fengycin competes with AIP for binding to *agrC*.	[21]
* Lb. helveticus *	Biosurfactant	*Staph. aureus*	In vitro	Inhibition of biofilm formation by interfering with AI-2 signaling and biofilm-related genes expression (*dltB, sarA, agrA, and icdA*).	[22]
			In vivo	Prevention of hemolytic activity through biofilm formation inhibition.	
*Lb. plantarum* KCTC10887BP	LPA	*Staph. aureus*	In vitro	Biofilm formation was inhibited. LPA induced AI-2 release in *Staph. Aureus*, which repressed biofilm-related genes.	[23]
*Lb. acidophillus* 30SC	N/A	*E. coli* O157:H7 43894	In vivo	Inhibition of AI-2 synthesis and modulation of microbial gut community.	[24]
* Lb. ** rhamnosus * GG microcapsules	N/A	* E. coli *	In vitro	Repression of *lsrK* and *luxS* genes (disruption in AI-2/*luxS*-typeQS network).	[13]
*Lb. acidophilus* A4	EPS	*E. coli* O157:H7	In vitro	Repression levels of curli genes (*crl*, *csgA*, and *csgB*) and chemotaxis (*cheY*) related to biofilm formation.	[25]
*Bifidobacterium longum* ATCC15707	CE	*E. coli* O157:H7	In vitro	Inhibition of AI-2 activity and virulence gene expression (*NifU*, *DsbA*, and *FlgI*).	[26]
*Lb. acidophilus* A4	N/A	*E. coli* (EHEC)	In vitro	Downregulation of biofilm-related genes (*crl*, *csgA*, and *csgB*) and chemotaxis (*cheY*).	[27]
*Lb. brevis* 3M004	N/A	*P. aeruginosa* PA002 biofilm formation	In vitro	Degradation of AIs and repression of biofilm formation, pyocyanin, and polysaccharide synthesis-related genes (*lasA*, *lasB,* and *PhzAB*).	[28]
*Lb. casei* CRL 431 *Lb. acidophilus* CRL 730	DKPs	*P. aeruginosa*	In vitro	DKPs compete with AI for binding QS receptors.	[29]
*Lb. rhamnosus* GG	CFS	*P. aeruginosa*	In vitro	Inhibition of AHL synthesis.	[30]
*Lb. casei* PTCC 1608	Lyophilized postbiotics	*P. aeruginosa*	In vitro	Repression of QS genes controlling biofilm formation and virulence (*rhlI*, *rhlR*, and *pelf*), potentially due to organic acid content.	[31]
*B. subtilis BR4*	Stigmatellin Y	*P. aeruginosa (ATCC 27853)*	In vitro	Stigmatellin Y competes with PQS signal for binding with *PqsR* gene, and thus, *PqsR-PQS* QS pathway is disrupted.	[32]
*B. paralicheniformis* ZP1	Lactonase	*P. aeruginosa*	In vitro	Inhibition of biofilm formation due to AHL hydrolysis by lactonase.	[33]
*B. subtilis* KATMIRA1933	Subtilisin	*L. monocytogenes* biofilm formation *E. coli* biofilm formation	In vitro	Inhibition of proton motive forces and efflux pumps.	[34]
*Lb. plantarum* C2	N/A	*E. coli* DSM 30083 *Enterobacter aerogenes DSM 30053* *Yersinia enterolitica* DSM 4780 *Leuconostoc lactis 20202* *Ent. durans DSM 20633* *B. megaterium F6*	In vitro	Antibacterial activity by plantaricin produced through QS mechanism. (AI not measured).	[11]
*Lb. plantarum* KU200656	CFS	*Staph. aureus* *Listeria monocytogenes* *E. coli* *S.* Typhimurium	In vitro	Biofilm-related genes are downregulated by anti-biofilm activity. (Exact QSI mechanism is not measured).	[35]
*Lb.** kefiri * 8321 and 83113 * Lb. plantarum * 83114	CFS	*S.* Enteritidis 115	In vitro	Biofilm formation inhibition. (QSI mechanism not investigated).	[36]
*B. subtilis* KATMIRA1933 *B. amyloliquefaciens* B-1895	CE/CFS	*S.* (Thompson, Enteritidis phage type 4, and Hadar)	In vitro	Biofilm inhibition due to the subtilosin effect. AI/*luxS* QS pathway is necessary for biofilm formation.	[37]
* W. viridescens * WM33 *W. confusa* WM36 (LAB)	CFS	*S.* Typhi and *S.* Typhimurium	In vitro	Inhibition of AI-2 activity and biofilm formation.	[38]
*Lb. reuteri* PFS4 *Ent. faecium* PFS13 and PFS14.	CFS	*S.* Typhimurium and *S.* Enteritidis	In vitro	Inhibition of biofilm formation. (Mechanism not investigated).	[39]
* Lb. coryniformis * NA-3	EPS	*B. cereus* and *S.* Typhimurium	In vitro	Inhibition of biofilm formation. (Mechanism not investigated).	[40]
* B. subtilis * ZK3814	Fengycin and surfactin	*Ent. faecalis* OG1RF	In vitro	Inhibition of fsr system, which regulates expression of proteolytic activity related-genes (*gelE*/*sprE*).	[41]
*Pd. pentosaceus*	Crude biosurfactant	* B. subtilis **and**Staph. aureus* * P. aeruginosa*, *Staph. aureus*, and *E. coli*	In vitro	Anti-QS and anti-biofilm activity.	[42]
*Lb. curvatus BSF206 and Pd. pentosaceus AC1-2*	CFS	*Str. mutans*	In vitro	Biofilm formation inhibition by downregulation of related genes (*tfA*, *gtfB*, *ftf*, and *brpA*). (*Exact mechanism not known*).	[43]
* Lb. paragasseri * MJM60645	Crude extract	*Str. mutans*	In vitro	Downregulation of biofilm-associated genes (*gtfB*, *gtfC*, *gtfD*, *gbpB*, *brpA*, *spaP*, *ftf*, and *smu0630)* by iminosugar, a novel chemical compound produced.	[44]
*Lb. rhamnosus* GG	Biosurfactant	* Str. mutans *	In vitro	Anti-biofilm activity due to downregulation of biofilm-related genes (*gtfB/C* and *ftf*).	[45]
*Lb. plantarum* K41	N/A	* Str. mutans *	In vitro and in vivo	Inhibition of biofilm formation by inhibition of exopolysaccharide production.	[46]
*Lb. casei* MCJΔ1 (expressed with AHL-lactonase *AiiK* gene)	Lactonase	* Aeromonas hydrophila *	In vitro	Enzymatic QQ activity of lactonase.	[47]
*Lb. curvatus* B.67 and *Lb. plantarum* M.2	Postbiotics	*L. monocytogenes*	In vitro	Repression of biofilm-related genes (*flaA*, *fbp*, *agrA*, *prfA*, and *hlyA*).	[48]
*Lb. curvatus* CRL1579	Lactocin	*L. monocytogenes*	In vitro	QSI mechanism not investigated.	[49]
*B. subtilis*-9	N/A	*E. coli (ETEC), S.* Typhimurium, *Staph. aureus (MSRA)*	In vitro	Biofilm inhibition in a cell-to-cell contact manner. Biofilm-related genes were repressed in ETEC (*bssS, luxS, and ihfB*).	[50]
*Lb. paracasei* L10	CFS	*V. parahaemolyticus*	In vitro	Biofilm formation significantly inhibited. (Mechanism not investigated).	[51]
*Lb. plantarum* LRCC 5193	LPA	* Str. mutans, E. faecalis, and Str. Gordonii *	In vitro	Biofilm formation inhibition. (QSI mechanism is suggested but not investigated).	[52]
* Lb. kefiranofaciens * DD2	CFS	* Str. mutans * and *Str. sobrinus*	In vitro	Antibiofilm activity through repression biofilm-associated genes (*ftf, comDE, brpA, and vicR*).	[53]

*Lb*: *Lactobacillus*; *C*.: *Clostridium*; *Staph*.: *Staphylococcus*; *L*.: *Listeria*; *B*.: *Bacillus*; *P*.: *Pseudomonas*; *Ent*.: *Enterococcus*; *Str*.: *Streptococcus*; Pd.: Pediococcus; *V*.: *Vibrio*; *W*.: *Weissella*; CE: cell extract; CFS: cell-free supernatant; LPA: lipoteichoic acid; N/A: not available; EPS: exopolysaccharides; SaeRS: two-component signaling system; DKP: diketopiperazine; AI: autoinducer; QQ: quorum quenching; PQS: Pseudomonas quinolone signal; BIC: biofilm inhibitory compound; QSI: QS inhibitor.

**Table 2 microorganisms-11-00793-t002:** Probiotics involved in the inhibition of QS mechanisms of food spoilage bacteria.

Microorganism	QSI	Target	Type of Study	Mechanism	Reference
*Lb. plantarum* ss-128	N/A	Spoilage bacterial community (*Shewanella*, *Carnobacterium*, and *Vagococcus)*	Food matrix (shrimp)	AI-2/*LuxS*-type QS system promotes growth of *Lb. plantarum* that reduces PH, protease activity, and growth of spoilage microorganism.	[81]
*B.* sp. AI96	Lactonase	*Aeromonas veronii* LP-11	In vitro	Spoilage inhibition due to QS disruption by lactonase.	[82]
*B. amyloliquefaciens* SBF1	Culture extract	* P. aeruginosa * PAO1	In vitro	Biofilm inhibition due to anti-QS activity.	[83]
*Lysinibacillus* sp. Gs50	Lactonase	*Pectobacterium carotovorum*	Food matrix	Vegetables soft rot reduced due to QQ of AHLs.	[84]
*B. cereus* RC1	DKPs	*Lelliottia amnigena*	Food matrix (vegetables)	Soft rot decreased due to anti-QS activity. DKPs compete for binding QS receptors with the AI.	[85]

**Table 3 microorganisms-11-00793-t003:** Association of QS molecules with gut health.

QS Molecule	Source	Type of Study	Health Outcomes	Reference
3-oxo-C12:2-HSL	Synthetic	In vitro	Anti-inflammatory effect.	[86]
3-oxo-C12:2 HSL	Human gut microbiota	Cross sectional (analysis of fecal samples of patients with IBDs)	Positive correlation with normobiosis (increased levels of Firmicutes).	[87]
		In vitro	Anti-inflammatory and positive effect on gut epithelial cell function.	
AI-2	Mutant *E. coli* engineered to overproduce AI-2	Animal study	Increased ratio of Firmicutes to Bacteroidetes in antibiotic-treated mice group.	[88]
AI-2	*Lb. rhamnosus* GG (LGG)	Animal study	Protective effect of Δ*luxS* LGG on intestinal cells is significantly lower than effect of wild-type LGG.	[89]
AI-2	Exogenous AI-2 added to milk	Animal study	Dysbiosis was reversed, and inflammation was ameliorated.	[90]
DPD (precursor of AI-2)	Exogenous (synthetic)	In vitro (co-culture of WCE of *Prevotella. intermedia*, *Prevotella nigrescens*, and estradiol with HMK cells)	DPD modulated the pro-inflammatory effect of estradiol + and inhibited biofilm formation.	[91]
AI-2	* Fusobacterium nucleatum *	Cross-sectional (analysis of fecal, saliva, and serum samples from patients with CRC and healthy people)	AI-2 levels are higher in CRC samples compared with control samples.	[92]
AI-2	Non-pathogenic * E. coli * BL21 and W3110	In vitro (co-culture with HCT-cells)	Increased expression of pro-inflammatory cytokine IL-8 but downregulated after 24 h.	[93]

IBDs: inflammatory bowel diseases; HMK: human gingival keratinocytes; HCT: human colon cancer; CRC: colorectal cancer; WCE: whole cell extract; DPD: dihydroxy-2,3-pentanedione.

## Data Availability

Data is contained within the article.

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
