# Peer review of "Anti-Quorum Sensing Activity of Probiotics: The Mechanism and Role in Food and Gut Health"

_microorganisms, 2023, doi:10.3390/microorganisms11030793_

Round 1

Reviewer 1 Report

Very good review of antiquorum sensing activity focused on food.

The bibliographic resource deployed is extensive and recent. 

It will be of great help to the scientific community.

It deserves to be published.

Author Response

Thank you so much for your appreciation of the manuscript.

Reviewer 2 Report

This review describes the current state of knowledge regarding probiotics with QS inhibitory activity against foodborne-pathogenic and spoilage bacteria. Overall, I believe that the topic can eventually be a meaningful contribution. However, the story of the manuscript is difficult to follow since it is not well summarised and clear. In my opinion, authors should try to elaborate the sections more, and summarise and explain the key concepts to not confuse the audience. 

Major comments:

The “Abstract” and the “Introduction” sections present disconnected sentences and slightly confusing concepts. It is unclear if the authors would like to focus on QSI in general, or on the QQ processes in particular. First, the authors seem to describe the QS inhibition, but they introduce the quorum quenching (QQ), without defining it (e.g. “Quorum quenching” and “secretome” in the “Abstract” section). However, they come back throughout the manuscript to the general processes of QSI. QQ is only a kind of QS inhibition (they are QSI processes through enzymatic activity, according to most of the literature). They should take this into account in order not to confuse the readers throughout the story. The authors also describe QS systems in Gram-negative and Gram-positive bacteria. But then, they focus on defining QSI for Gram-negative bacteria (Lines 71-75, 347, 426-428, among others), and then they talk about QSI in both, which made the manuscript difficult to read. It is hard to follow the story without getting lost even if the reader is an expert in QSI. 

In general, the following sections need more elaboration, as they present the same problems as the previous section. The general idea of the manuscript is great and would be interesting for the audience, however, the authors should try to improve the wording of the text. The wording and spelling are not good, there are some sentences in italics, and some bacterial species without italics. The names of bacteria should be completely defined the first time. There are some underlined words (e.g. Table 1) as well as the authors use contractions. Do not use contractions in formal language. Genes and operons should be italicised. “SDiA” should be “SdiA”. Better writing and good spelling would improve the manuscript and help the audience to read it. Review the manuscript in detail to avoid these errors. A better reorganisation of the information and conclusions would clarify the story and strongly improve the manuscript to make it more attractive.  

Table 1: It is not clear if these results have been demonstrated as probiotics or only as microorganisms/cell extracts/enzymes with QS inhibition activity. 

Line 110: what databases have they used? 

- Define the complete names of bacterial species the first time they write them. e.g. L. reuteri .

- The addition of another column in the first position in the table showing the name of the microorganism (in alphabetic order), would clarify the reading. In the second column would be the QS inhibitor (if it is an enzyme, a cellular extract or the bacterial cells). If authors prefer to order the target microorganism in alphabetic order, they should change this column to the first position in the table.

- Try to unify the information on the table. E.g. some bacteria have the strain included, and some do not. 

- “lyophilized postbiotics from L. casei”. Postbiotics?

Table 2. The same is applied to table 1. The first column would be clear if it was divided. When the authors include QS inhibitors with in-vitro activity demonstrated, why do they choose these articles? There are many QS inhibitors against P. aeruginosa for example.

Section 3. A summary table for this section would be great.

Section 4. This section is so interesting. However, it is placed as the last section and the authors only found a few articles. 

Minor comments:

Line 26: the word “species” or “specific” is lacking in “inter- and intra- communication” 

Lines 32-34: Not all Gram-negative bacteria present the canonical luxI/luxR-type QS system. Reflect it on the text. The same applies to Gram-positive bacteria. If authors do not introduce all the details of all QS systems they should talk “in general”.

Line 41: Are there studies about probiotics with the AI-3 system?

Line 52: something is missing in “pectinolytic, lipolytic and proteolytic), reword.

Line 52-56: Does this happen exactly through QS processes? Or are there more virulence factors involved? I would change “could” instead of “can”. Some sentences are a little pretentious, e.g. “can” instead of “could” or “acts” in line 57. Although the AHL presence in milk has the potential to deteriorate food, is it completely true that bacteria deteriorate milk only through QS? 

Line 55: define “HSL/AI1”.

Line 58: add a reference.

Line 74: Are they only lactonase enzymes as QSI? In other sections, authors describe more enzyme activities. They should present a general idea at the beginning.

Line 92: “AGR” should not be in italics.

Line 94: Define “QQ” in the text. Why do the authors introduce the concept of QQ (enzymatic QSI) if they are going to describe all the QSI mechanisms?

Lines 101,102,103 and 104: References are missing.

Line 117: Define “EPS” and “DPK” in the text.

Line 126: “SDiA” should be “SdiA”. 

Line 131: Does “binding of QSI with SdiA amino acids” means binding the QSI with the SdiA regulator? Reword it.

Line 135, 162-167: in italics.

Lines 142-144: reword it.

Lines 136-137: What do the authors believe this role of the EPS might be?

Line 133: Authors wrote: “QQ compounds that exert the anti-QS

effect aren’t yet determined.” But then, they talk about QQ against E. coli in lines 160-162. Clarify it.

Line 170: Define “LPA”.

Line 190: name of genes, and operons should be in italics: Bacillus, L. brevis…Review the manuscript.

Line 216: is not this section for foodborne pathogens and not spoilage pathogens? Also, why the authors included G. vaginalis and S. pneumoniae as foodborne pathogens?

Lines 244-246. Something needs to be added to the sentences. Reword them.

Line 248. Do the authors know any examples of this interaction?

Line 271: “LuxS/AHL” ?

Lines 282-287: What do the authors think about the limitation of the use of probiotics in food? Do they know if probiotics could alter the properties or quality of food? Also, what do they think about the interference of probiotics with commensal bacteria in the gut for example?

Lines 297-298: Add a reference.

Line 306: Add a reference. Why do the authors say OC12-HLS is the most studied AHL?

Lines 313-314. Reword the sentence, and, what is this study? 

Lines 342-346. Reword these sentences. What do the authors mean by “QQ such as carbapenems and azithromycin”? 

Lines 405-408: It needs to be clarified the idea of these sentences. What do the authors mean by this study?

Author Response

In black your comments, our notes in red.

This review describes the current state of knowledge regarding probiotics with QS inhibitory activity against foodborne-pathogenic and spoilage bacteria. Overall, I believe that the topic can eventually be a meaningful contribution. However, the story of the manuscript is difficult to follow since it is not well summarised and clear. In my opinion, authors should try to elaborate the sections more, and summarise and explain the key concepts to not confuse the audience. 

We appreciate the effort you took to thoroughly review our paper and provide such thoughtful and detailed feedback. Your comments have helped us so much to identify the parts we improved. The manuscript has been revised to elaborate the main aims of the review.

The “Abstract” and the “Introduction” sections present disconnected sentences and slightly confusing concepts. It is unclear if the authors would like to focus on QSI in general, or on the QQ processes in particular. First, the authors seem to describe the QS inhibition, but they introduce the quorum quenching (QQ), without defining it (e.g. “Quorum quenching” and “secretome” in the “Abstract” section). However, they come back throughout the manuscript to the general processes of QSI. QQ is only a kind of QS inhibition (they are QSI processes through enzymatic activity, according to most of the literature). They should take this into account in order not to confuse the readers throughout the story. The authors also describe QS systems in Gram-negative and Gram-positive bacteria. But then, they focus on defining QSI for Gram-negative bacteria (Lines 71-75, 347, 426-428, among others), and then they talk about QSI in both, which made the manuscript difficult to read. It is hard to follow the story without getting lost even if the reader is an expert in QSI. 

We revised Abstract and Introduction sections accordingly. We clarify and clearly state that QQ (QS inhibition by enzymes) is a part of QS inhibition process. We rewrote some parts to make the text more clear and more fluent to the reader.

In general, the following sections need more elaboration, as they present the same problems as the previous section. The general idea of the manuscript is great and would be interesting for the audience, however, the authors should try to improve the wording of the text. The wording and spelling are not good, there are some sentences in italics, and some bacterial species without italics. The names of bacteria should be completely defined the first time. There are some underlined words (e.g. Table 1) as well as the authors use contractions. Do not use contractions in formal language. Genes and operons should be italicised. “SDiA” should be “SdiA”. Better writing and good spelling would improve the manuscript and help the audience to read it. Review the manuscript in detail to avoid these errors. A better reorganisation of the information and conclusions would clarify the story and strongly improve the manuscript to make it more attractive.

We modified all other sctions accordingly.

Table 1: It is not clear if these results have been demonstrated as probiotics or only as microorganisms/cell extracts/enzymes with QS inhibition activity. 

We have reorganised the Table 1 to make more clear the microorganisms and the QSIs involved in the study.

Line 110: what databases have they used? 

We do not mention anymore databases.

Define the complete names of bacterial species the first time they write them. e.g. L. reuteri .

We report abbreviations of microorganisms in the footnote of Table 1. We think it should work well for the reader.

The addition of another column in the first position in the table showing the name of the microorganism (in alphabetic order), would clarify the reading. In the second column would be the QS inhibitor (if it is an enzyme, a cellular extract or the bacterial cells). If authors prefer to order the target microorganism in alphabetic order, they should change this column to the first position in the table.

We followed your suggestion to add a new column as reported in the above note. Very hard to organize in alphabetic order. We organize according to the the story we want to tell in the text.

Try to unify the information on the table. E.g. some bacteria have the strain included, and some do not.

We didn't include the strain where it was not mentioned in the reference.

"lyophilized postbiotics from L. casei”. Postbiotics?

Yes, exactly, the microorganism was used in this study as a postbiotic, that is without active cells.

Table 2. The same is applied to table 1. The first column would be clear if it was divided. When the authors include QS inhibitors with in-vitro activity demonstrated, why do they choose these articles? There are many QS inhibitors against P. aeruginosa for example.

We changed the Table 2 and we removed all the references in which the QS inhibition is not related to potential probiotic microorganisms.

Section 3. A summary table for this section would be great.

Done.

Section 4. This section is so interesting. However, it is placed as the last section and the authors only found a few articles.

It's not yet well investigated field, but we are actually going to work on this.

Minor comments

We followed all your suggestions.

Reviewer 3 Report

The manuscript "Anti-Quorum Sensing Activity of Probiotics: The Mechanism and Role in Food and Gut Health" has an interesting outcome. However, for publication in Microorganisms, the manuscript needs to be improved.

Many grammatically problematic sentences were found throughout the manuscript, which must be checked and corrected precisely. A professional English editing service is highly recommended.

  1. L13, 14, Table 1, and so on: Should be italic “in vitro”. Check and make corrections throughout the manuscript.
  2. L15: Put comma “cytokines response, gut dysbiosis, and….”
  3. L27: Put reference https://doi.org/10.1016/j.tifs.2020.03.019
  4. L52: Put comma “pectinolytic, lipolytic, and proteolytic”
  5. L62-65: Probiotic microorganisms have their own QS molecules to modulate their efficacy. In your sentence, do you mean probiotics inhibit their own QS molecules? Rephrase the sentences.
  6. L70: Put comma “compounds, products, and bacteria”
  7. L71: biofilm formation by
  8. L106: Delete “In Error! Reference source not found”
  9. Table 1 and so on: Use the full form of bacteria when used for the first time throughout the manuscript. Make corrections throughout the manuscript.
  10. L134, 236, 334, and so on: Why references are bold?
  11. L258, 260, 296, and so on: Rephrase the sentences
  12. L288: Put comma “QS, Biofilm Formation, and Gut Health”
  13. Many spacing and punctuation marks problems are found throughout the manuscript. Revision required.

Author Response

In black your comments, in red our notes.

The manuscript "Anti-Quorum Sensing Activity of Probiotics: The Mechanism and Role in Food and Gut Health" has an interesting outcome. However, for publication in Microorganisms, the manuscript needs to be improved.

Thank you very much for your suggestions helped us to enhance the general quality of the manuscript.

Many grammatically problematic sentences were found throughout the manuscript, which must be checked and corrected precisely. A professional English editing service is highly recommended.

We totally revised the manuscript for grammar and a professional English editing was performed.

L13, 14, Table 1, and so on: Should be italic “in vitro”. Check and make corrections throughout the manuscript.

Done.

L15: Put comma “cytokines response, gut dysbiosis, and….”

We changed the sentence.

L27: Put reference https://doi.org/10.1016/j.tifs.2020.03.019

We confined the review just to the papers in which the QS inhibition role of probiotics is mentioned.

L52: Put comma “pectinolytic, lipolytic, and proteolytic”

Done.

L62-65: Probiotic microorganisms have their own QS molecules to modulate their efficacy. In your sentence, do you mean probiotics inhibit their own QS molecules? Rephrase the sentences.

Done.

Put comma “compounds, products, and bacteria”

Done.

L71: biofilm formation by

Done.

L106: Delete “In Error! Reference source not found”

From the Journal editing.

Table 1 and so on: Use the full form of bacteria when used for the first time throughout the manuscript. Make corrections throughout the manuscript.

We added abbreviations of microorganisms in the footnotes of the table.

L134, 236, 334, and so on: Why references are bold?

We fixed.

L258, 260, 296, and so on: Rephrase the sentences

Done.

L288: Put comma “QS, Biofilm Formation, and Gut Health”

Done.

Many spacing and punctuation marks problems are found throughout the manuscript. Revision required.

Done.

Round 2

Reviewer 2 Report

Dear authors,

Thank you for considering all the different suggestions. The manuscript has been improved. However, I recommend that authors use a text checker or a language corrector tool, since there are several spelling and writing mistakes that make the revision process more difficult.

Moreover, I would like to add a few comments for the authors:

Line 61, 96: I would write “luxRI-type QS system” or add the word “homologous” since this is the canonical system in Gram-negative bacteria but not all of present the luxR or luxI exactly. Revise the manuscript taking this into account.

Line 81: Not all the QQ enzymes, as the authors explain below, “hydrolyze the lactone bond in AHL”. Add “i.e” before this statement or explain other mechanisms.

Lines 143-145: Please, add a reference.

Line 153: Rephrase this sentence.

Line 156: Please, add a reference.

Lines 169-182: In this paragraph the authors only explain in detail the effect of biosurfactants on S. aureus. What is the effect on the other bacteria that they have mentioned above (Lines 168-169)?

Line 214: “gene”

Lines 229-230: Please reword the sentence.

Lines 250-251: What do the authors think about how probiotics can affect food properties?

Lines 252-296: This paragraph has many spelling and writing mistakes. Please, carefully revise your manuscript before submiting it.

Lines 281-283; 298-299: Please, reword these sentences.

Line 298: This reference is not found.

210, 359: rhl, luxS. All genes should be in italics. Please review your manuscript in detail.

Line 302: “in acidity” where? So what is the conclusion?

Lines 317-319: Please, reword the sentence.

Tables: the description of abbreviations in tables can be all together below the tables. i.e. Table 3, DPD description.

Line 366. Helicobacter?

Line 385. As furanone has been described as a toxic compound, do the authors know about the toxicity of other QS inhibitors they mention throughout their manuscript?

Line 390-392: What do the authors mean by “to develop resistance to antibiotic QSIs carbapenems and azithromycin”? If azithromycin has been described as having QSI properties, it is not clear in the manuscript.

Or does it mean that they can develop resistance to QSI mediated by carbapenems and azithromycin? Please, clarify it.

Lines 406-408 If AI-2 can be used as a marker for CRC, how the authors believe that AI-2 supplementation will restore the anti-tumor response?

Lines 442-443: Please, reword the sentence.

Lines 455. Please, explain what “protein expression of GG” means to the audience. What type of proteins or functions do they refer to?

Author Response

In black teh reviewer comment, in red the author answers.

Line 61, 96: I would write “luxRI-type QS system” or add the word “homologous” since this is the canonical system in Gram-negative bacteria but not all of present the luxR or luxI exactly. Revise the manuscript taking this into account.

Done

Line 81: Not all the QQ enzymes, as the authors explain below, “hydrolyze the lactone bond in AHL”. Add “i.e” before this statement or explain other mechanisms.

We add: (e.g. hydrolysis of lactone bond in AHL)

Lines 143-145: Please, add a reference.

Done

Line 153: Rephrase this sentence.

Done

Line 156: Please, add a reference.

Done

Lines 169-182: In this paragraph the authors only explain in detail the effect of biosurfactants on S. aureus. What is the effect on the other bacteria that they have mentioned above (Lines 168-169)?

We rephrased the sentence to clarify.

Line 214: “gene”

Done

Lines 229-230: Please reword the sentence.

Done

Lines 250-251: What do the authors think about how probiotics can affect food properties?

Interesting question. We are currently working on the "metabolic attenuation" of probiotics for food fuctionalization with probiotics. We published the paper Application of ultrasound and microencapsulation on Limosilactobacillus reuteri DSM 17938 as a metabolic attenuation strategy for tomato juice probiotication. https://doi.org/10.1016/j.heliyon.2022.e10969

Lines 252-296: This paragraph has many spelling and writing mistakes. Please, carefully revise your manuscript before submiting it.

Yes, you are right. We checked, corrected and rephrased all the paragraph 2.2

Lines 281-283; 298-299: Please, reword these sentences.

Done

Line 298: This reference is not found.

We was wrong to indicate table 1. We changed with table 2.

210, 359: rhl, luxS. All genes should be in italics. Please review your manuscript in detail.

Done

Line 302: “in acidity” where? So what is the conclusion?

We changed.

Lines 317-319: Please, reword the sentence.

Done

Tables: the description of abbreviations in tables can be all together below the tables. i.e. Table 3, DPD description.

Done

Line 366. Helicobacter?

Yes, correct, thank you.

Line 385. As furanone has been described as a toxic compound, do the authors know about the toxicity of other QS inhibitors they mention throughout their manuscript?

At the best of our knowledge EFSA showed safe concerns on furan, not on the furanone. In any case no QSI has been reported as toxic.

Line 390-392: What do the authors mean by “to develop resistance to antibiotic QSIs carbapenems and azithromycin”? If azithromycin has been described as having QSI properties, it is not clear in the manuscript. Or does it mean that they can develop resistance to QSI mediated by carbapenems and azithromycin? Please, clarify it.

We clarified.

Lines 406-408 If AI-2 can be used as a marker for CRC, how the authors believe that AI-2 supplementation will restore the anti-tumor response?

The paragraph is now slightly reworded for more clarification.

Lines 442-443: Please, reword the sentence.

Done

Lines 455. Please, explain what “protein expression of GG” means to the audience. What type of proteins or functions do they refer to?

We rewrote the sentence.

Reviewer 3 Report

Accept in the present form. Thank you.

Author Response

Thank you.